# Geometry Based Data Generation

**Ofir Lindenbaum**[*]
Applied Mathematics Program
Yale University
New Haven, CT 06511
`ofir.lindenbaum@yale.edu`

**Jay S. Stanley III**[*]
Computational Biology & Bioinformatics Program
Yale University
New Haven, CT 06510
`jay.stanley@yale.edu`

**Guy Wolf**[†]
Applied Mathematics Program
Yale University
New Haven, CT 06511
`guy.wolf@yale.edu`

**Smita Krishnaswamy**[†] ✉
Departments of Genetics & Computer Science
Yale University
New Haven, CT 06510
`smita.krishnawamy@yale.edu`

## Abstract

We propose a new type of generative model for high-dimensional data that learns a manifold geometry of the data, rather than density, and can generate points evenly along this manifold. This is in contrast to existing generative models that represent data density, and are strongly affected by noise and other artifacts of data collection. We demonstrate how this approach corrects sampling biases and artifacts, thus improves several downstream data analysis tasks, such as clustering and classification. Finally, we demonstrate that this approach is especially useful in biology where, despite the advent of single-cell technologies, rare subpopulations and gene-interaction relationships are affected by biased sampling. We show that SUGAR can generate hypothetical populations, and it is able to reveal intrinsic patterns and mutual-information relationships between genes on a single-cell RNA sequencing dataset of hematopoiesis.

## 1 Introduction

Manifold learning methods in general, and diffusion geometry ones in particular (Coifman & Lafon, 2006), are traditionally used to infer latent representations that capture intrinsic geometry in data, but they do not relate them to original data features. Here, we propose a novel data synthesis method, which we call SUGAR (Synthesis Using Geometrically Aligned Random-walks), for generating data in its original feature space while following its intrinsic geometry. This geometry is inferred by a diffusion kernel that captures a data-driven manifold and reveals underlying structure in the full range of the data space – including undersampled regions that can be augmented by new synthesized data. Geometry-based data generation with SUGAR is motivated by numerous uses in data exploration. For instance, in biology, despite the advent of single-cell technologies such as single-cell RNA sequencing and mass cytometry, sampling biases and artifacts often make it difficult to evenly sample the data space. Rare populations of relevance to disease and development are often left out (Grün et al., 2015). By learning the data geometry rather than density, SUGAR is able to generate hypothetical cell types for exploration, and uncover patterns and interactions in the data.

Further, imbalanced data is problematic for many machine learning applications. In classification, for example, class density can strongly bias some classifiers (He & Garcia, 2009; López et al.,

---

[*]These authors contributed equally
[†]These authors contributed equally; ✉ Corresponding author

2013; Hensman & Masko, 2015). In clustering, imbalanced sampling of ground truth clusters can lead to distortions in learned clusters (Xuan et al., 2013; Wu, 2012). Sampling biases can also corrupt regression tasks; relationship measures such as mutual information are heavily weighted by density estimates and thus may mis-quantify the strength of dependencies with data whose density is concentrated in a particular region of the relationship (Krishnaswamy et al., 2014). SUGAR can aid such machine learning algorithms by generating data that is balanced along its manifold.

There are several advantages of our approach over contemporary generative models. Most other generative models attempt to learn and replicate the density of the data; this approach is intractable in high dimensions. Distribution-based generative models typically require vast simplifications such as parametric forms or restriction to marginals in order to become tractable. Examples for such methods include Gaussian Mixture Models (GMM, Rasmussen, 2000), variational Bayesian methods (Beal & Ghahramani, 2003), and kernel density estimates (Scott, 2008). In contrast to these methods, SUGAR does not rely on high dimensional probability distributions or parametric forms. SUGAR selectively generates points to equalize density; as such, the method can be used generally to compensate for sparsity and heavily biased sampling in data in a way that is agnostic to downstream application. In other words, whereas more specialized methods may use prior information (e.g., labels) to correct class imbalances for classifier training (Chawla et al., 2002), SUGAR does not require such information and can apply even in cases such as clustering or regression, where such information does not exist.

Here, we construct SUGAR from diffusion methods and theoretically justify its density equalization properties. We then demonstrate SUGAR on imbalanced artificial data. Subsequently, we use SUGAR to improve classification accuracy on 61 imbalanced datasets. We then provide an illustrative synthetic example of clustering with SUGAR and show the clustering performance of the method on 115 imbalanced datasets obtained from the KEEL-dataset repository (Alcalá-Fdez et al., 2009). Finally, we use SUGAR for exploratory analysis of a biological dataset, recovering imbalanced cell types and restoring canonical gene-gene relationships.

## 2   Related Work

Most existing methods for data generation assume a probabilistic data model. Parametric density estimation methods, such as Rasmussen (2000) or Varanasi & Aazhang (1989), find a best fitting parametric model for the data using maximum likelihood, which is then used to generate new data. Nonparametric density estimators (e.g., Seaman & Powell, 1996; Scott, 1985; Giné & Guillou, 2002) use a histogram or a kernel (Scott, 2008) to estimate the generating distribution. Recently, Variational Auto-Encoders (VAE, Kingma & Welling, 2014; Doersch, 2016) and Generative Adversarial Networks (GAN, Goodfellow et al., 2014) have been demonstrated for generating new points from complex high dimensional distributions.

A family of manifold based Parzen window estimators are presented in Vincent & Bengio (2003); Bengio & Monperrus (2005); Bengio et al. (2006). These methods exploit manifold structures to improve density estimation of high dimensional data. Markov-Chain Monte Carlo (MCMC) on implicitly defined manifolds was presented in Girolami & Calderhead (2011); Brubaker et al. (2012). There, the authors use implicit constraints to generate new points that follow a manifold structure. Another scheme by Öztireli et al. (2010) defines a spectral measure to resample existing points such that manifold structure is preserved and the density of points is uniform. These methods differ from the proposed approach as they either require implicit constraints or they change the values of existing points in the resampling process.

## 3   Background

### 3.1   Diffusion Geometry

Coifman & Lafon (2006) proposed the nonlinear dimensionality reduction framework called Diffusion Maps (DM). This popular method robustly captures an intrinsic manifold geometry using a row-stochastic Markov matrix associated with a graph of the data. This graph is commonly constructed

using a Gaussian kernel

$$\mathcal{K}(\boldsymbol{x}_i, \boldsymbol{x}_j) \triangleq K_{i,j} = \exp\left(-\frac{\|\boldsymbol{x}_i - \boldsymbol{x}_j\|^2}{2\sigma^2}\right), \quad i, j = 1, ..., N \tag{1}$$

where $x_1, \ldots, x_N$ are data points, and $\sigma$ is a bandwidth parameter that controls neighborhood sizes. Then, a diffusion operator is defined as the row-stochastic matrix $P_{i,j} = \mathcal{P}(\boldsymbol{x}_i, \boldsymbol{x}_j) = [\boldsymbol{D}^{-1}\boldsymbol{K}]_{i,j}$, $i, j = 1, ..., N$, where $\boldsymbol{D}$ is a diagonal matrix with values corresponding to the degree of the kernel $D_{i,i} = \hat{d}(i) = \sum_j \mathcal{K}(\boldsymbol{x}_i, \boldsymbol{x}_j)$. The degree $\hat{d}(i)$ of each point $\boldsymbol{x}_i$ encodes the total connectivity the point has to its neighbors. The Markov matrix $\boldsymbol{P}$ defines an imposed diffusion process, shown by Coifman & Lafon (2006) to efficiently capture the diffusion geometry of a manifold $\mathcal{M}$.

The DM framework may be used for dimensionality reduction to embed data using the eigendecomposition of the diffusion operator. However, in this paper, we do not directly use the DM embedding, but rather a variant of the operator $\boldsymbol{P}$ that captures diffusion geometry. In Sec. 4, we explain how this operator allows us to ensure the data we generate follows diffusion geometry and the manifold structure it represents.

### 3.2 Measure Based Gaussian Correlation

Bermanis et al. (2016a,b) suggest the Measure-based Gaussian Correlation (MGC) kernel as an alternative to the Gaussian kernel (Eq. 1) for constructing diffusion geometry based on a measure $\mu$. The measure could be provided in advance or approximated based on the data samples. The MGC kernel with a measure $\mu(\boldsymbol{r})$, $\boldsymbol{r} \in \boldsymbol{X}$, defined over a set $\boldsymbol{X}$ of reference points, is

$$\hat{\mathcal{K}}(\boldsymbol{x}_i, \boldsymbol{x}_j) = \sum_{\boldsymbol{r} \in \boldsymbol{X}} \mathcal{K}(\boldsymbol{x}_i, \boldsymbol{r})\mathcal{K}(\boldsymbol{r}, \boldsymbol{x}_j)\mu(\boldsymbol{r}), \quad i, j = 1, ..., N,$$

where the kernel $\mathcal{K}$ is some decaying symmetric function. Here, we use a Gaussian kernel for $\mathcal{K}$ and a sparsity-based measure for $\mu$.

### 3.3 Kernel Bandwidth Selection

The choice of kernel bandwidth $\sigma$ in Eq. 1 is crucial for the performance of Gaussian-kernel methods. For small values of $\sigma$, the resulting kernel $\mathcal{K}$ converges to the identity matrix; inversely, large values of $\sigma$ yield the all-ones matrix. Many methods have been proposed for tuning $\sigma$. A range of values is suggested in Singer et al. (2009) based on an analysis of the sum of values in $\mathcal{K}$. Lindenbaum et al. (2017) presented a kernel scaling method that is well suited for classification and manifold learning tasks. We describe here two methods for setting the bandwidth: a global scale suggested in Keller et al. (2010) and an adaptive local scale based on Zelnik-Manor & Perona (2005).

For degree estimation we use the max-min bandwidth (Keller et al., 2010) as it is simple and effective. The max-min bandwidth is defined by

$$\sigma^2_{\text{MaxMin}} = \mathcal{C} \cdot \max_j[\min_{i, i \neq j}(\|\boldsymbol{x}_i - \boldsymbol{x}_j\|^2)],$$

where $\mathcal{C} \in [2, 3]$. This approach attempts to force each point to be connected to at least one other point. This method is simple, but highly sensitive to outliers. Zelnik-Manor & Perona (2005) propose adaptive bandwidth selection. At each point $\boldsymbol{x}_i$, the scale $\sigma_i$ is chosen as the $L^1$ distance of $\boldsymbol{x}_i$ from its $r$-th nearest neighbor. This adaptive bandwidth guarantees that at least half of the points are connected to $r$ neighbors. Since an adaptive bandwidth obscures density biases, it is more suitable for applying the resulting diffusion process to the data than for degree estimation.

## 4 Data Generation

### 4.1 Problem Formulation

Let $\mathcal{M}$ be a $d$ dimensional manifold that lies in a higher dimensional space $\mathbb{R}^D$, with $d < D$, and let $\boldsymbol{X} \subseteq \mathcal{M}$ be a dataset of $N = |\boldsymbol{X}|$ data points, denoted $\boldsymbol{x}_1, \ldots, \boldsymbol{x}_N$, sampled from the manifold. In this paper, we propose an approach that uses the samples in $\boldsymbol{X}$ in order to capture the manifold

---
**Algorithm 1** SUGAR: Synthesis Using Geometrically Aligned Random-walks
---
**Input:** Dataset $\boldsymbol{X} = \{\boldsymbol{x}_1, \boldsymbol{x}_2, \ldots, \boldsymbol{x}_N\}, \boldsymbol{x}_i \in \mathbb{R}^D$.
**Output:** Generated set of points $\boldsymbol{Y} = \{\boldsymbol{y}_1, \boldsymbol{y}_2, \ldots, \boldsymbol{y}_M\}, \boldsymbol{y}_i \in \mathbb{R}^D$.

1: Compute the diffusion geometry operators $\boldsymbol{K}, \boldsymbol{P}$, and degrees $\hat{d}(i), i = 1, ..., N$ (see Sec. 3)
2: Define a sparsity measure $\hat{s}(i), i = 1, ..., N$ (Eq. 2).
3: Estimate a local covariance $\boldsymbol{\Sigma}_i, i = 1, ..., N$, using $k$ nearest neighbors around each $\boldsymbol{x}_i$.
4: For each point $i = 1, ..., N$ draw $\hat{\ell}(i)$ vectors (see Sec. 4.3) from a Gaussian distribution $\mathcal{N}(\boldsymbol{x}_i, \boldsymbol{\Sigma}_i)$. Let $\hat{\boldsymbol{Y}}_0$ be a matrix with these $M = \sum_{i=1}^{N} \hat{\ell}(i)$ generated vectors as its rows.
5: Compute the sparsity based diffusion operator $\hat{\boldsymbol{P}}$ (see Sec 4.2).
6: Apply the operator $\hat{\boldsymbol{P}}$ at time instant $t$ to the new generated points in $\hat{\boldsymbol{Y}}_0$ to get diffused points as rows of $\boldsymbol{Y}_t = \hat{\boldsymbol{P}}^t \cdot \boldsymbol{Y}_0$.
7: Rescale $\boldsymbol{Y}_t$ to get the output $\boldsymbol{Y}[\cdot, j] = \boldsymbol{Y}_t[\cdot, j] \cdot \frac{\text{percentile}(\boldsymbol{X}[\cdot, j], .99)}{\max \boldsymbol{Y}_t[\cdot, j]}, j = 1, \ldots, D$, in order to fit the original range of feature values in the data.
---

geometry and generate new data points from the manifold. In particular, we focus on the case where the points in $\boldsymbol{X}$ are unevenly sampled from $\mathcal{M}$, and aim to generate a set of $M$ new data points $\boldsymbol{Y} = \{\boldsymbol{y}_1, ..., \boldsymbol{y}_M\} \subseteq \mathbb{R}^D$ such that 1. the new points $\boldsymbol{Y}$ approximately lie on the manifold $\mathcal{M}$, and 2. the distribution of points in the combined dataset $\boldsymbol{Z} \triangleq \boldsymbol{X} \cup \boldsymbol{Y}$ is uniform. Our proposed approach is based on using an intrinsic diffusion process to robustly capture a manifold geometry from $\boldsymbol{X}$ (see Sec. 3). Then, we use this diffusion process to generate new data points along the manifold geometry while adjusting their intrinsic distribution, as explained in the following sections.

## 4.2 SUGAR: Synthesis Using Geometrically Aligned Random-walks

SUGAR initializes by forming a Gaussian kernel $\boldsymbol{G_X}$ (see Eq. 1) over the input data $\boldsymbol{X}$ in order to estimate the degree $\hat{d}(i)$ of each $\boldsymbol{x}_i \in \boldsymbol{X}$. Because the space in which the degree is estimated impacts the output of SUGAR, $\boldsymbol{X}$ may consist of the full data dimensions or learned dimensions from manifold learning algorithms. We then define the sparsity of each point $\hat{s}(i)$ via

$$\hat{s}(i) \triangleq [\hat{d}(i)]^{-1}, i = 1, ..., N. \tag{2}$$

Subsequently, we sample $\hat{\ell}(i)$ points $\boldsymbol{h}_j \in \boldsymbol{H}_i, j = 1, ..., \hat{\ell}(i)$ around each $\boldsymbol{x}_i \in \boldsymbol{X}$ from a set of localized Gaussian distributions $G_i = \mathcal{N}(\boldsymbol{x}_i, \boldsymbol{\Sigma}_i) \in \mathcal{G}$. The choice of $\hat{\ell}(i)$ based on the density (or sparsity) around $\boldsymbol{x}_i$ is discussed in Sec. 4.3. This construction elaborates local manifold structure in meaningful directions by 1. compensating for data sparsity according to $\hat{s}(i)$, and 2. centering each $G_i$ on an existing point $\boldsymbol{x}_i$ with local covariance $\boldsymbol{\Sigma}_i$ based on the $k$ nearest neighbors of $\boldsymbol{x}_i$. The set of all $M = \sum_i \hat{\ell}(i)$ new points, $\boldsymbol{Y}_0 = \{\boldsymbol{y}_1, ..., \boldsymbol{y}_M\}$, is then given by the union of all local point sets $\boldsymbol{Y}_0 = \boldsymbol{H}_1 \cup \boldsymbol{H}_2 \cup ... \cup \boldsymbol{H}_N$. Next, we construct a sparsity-based MGC kernel (see Sec. 3.2)

$$\hat{\mathcal{K}}(\boldsymbol{y}_i, \boldsymbol{y}_j) = \sum_r \mathcal{K}(\boldsymbol{y}_i, \boldsymbol{x}_r) \mathcal{K}(\boldsymbol{x}_r, \boldsymbol{y}_j) \hat{s}(r)$$

using the affinities in the sampled set $\boldsymbol{X}$ and the generated set $\boldsymbol{Y}_0$. We use this kernel to pull the new points $\boldsymbol{Y}_0$ toward the sparse regions of the manifold $\mathcal{M}$ using the row-stochastic diffusion operator $\hat{\boldsymbol{P}}$ (see Sec. 3.1). We then apply the powered operator $\hat{\boldsymbol{P}}^t$ to $\boldsymbol{Y}_0$, which averages points in $\boldsymbol{Y}_0$ according to their neighbors in $\boldsymbol{X}$.

The powered operator $\hat{\boldsymbol{P}}^t$ controls the *diffusion distance* over which points are averaged; higher values of $t$ lead to wider local averages over the manifold. The operator may be modeled as a low pass filter in which higher powers decrease the cutoff frequency. Because $\boldsymbol{Y}_0$ is inherently noisy in the ambient space of the data, $\boldsymbol{Y}_t = \hat{\boldsymbol{P}}^t \cdot \boldsymbol{Y}_0$ is a denoised version of $\boldsymbol{Y}_0$ along $\mathcal{M}$. The number of steps required can be set manually or using the Von Neumann Entropy as suggested by Moon et al. (2017b). Because the filter $\hat{\boldsymbol{P}}^t \cdot \boldsymbol{Y}_0$ is not power preserving, $\boldsymbol{Y}_t$ is rescaled to fit the range of original values of $\boldsymbol{X}$. A full description of the approach is given in Alg. 1.

### 4.3 Manifold Density Equalization

The generation level $\hat{\ell}(i)$ in Alg. 1 (step 4), i.e., the amount of points generated around each $\boldsymbol{x}_i$, determines the distribution of points in $\boldsymbol{Y}_1$. Given a biased dataset $\boldsymbol{X}$, we wish to generate points in sparse regions such that the resulting density over $\mathcal{M}$ becomes uniform. To do this we have proposed to draw $\hat{\ell}(i)$ points around each point $\boldsymbol{x}_i, i = 1, ..., N$, from $\mathcal{N}(\boldsymbol{x}_i, \boldsymbol{\Sigma}_i)$ (as described in Alg. 1). The following proposition provides bounds on the "correct" number of points $\hat{\ell}(i), i = 1, ..., N$, required to balance the density over the manifold by equalizing the degrees $\hat{d}(i)$.

**Proposition 4.1.** *The generation level $\hat{\ell}(i)$ required to equalize the degree $\hat{d}(i)$, is bounded by*

$$\det\left(\boldsymbol{I} + \frac{\boldsymbol{\Sigma}_i}{2\sigma^2}\right)^{\frac{1}{2}} \frac{\max(\hat{d}(\cdot)) - \hat{d}(i)}{\hat{d}(i) + 1} - 1 \leq \hat{\ell}(i) \leq \det\left(\boldsymbol{I} + \frac{\boldsymbol{\Sigma}_i}{2\sigma^2}\right)^{\frac{1}{2}} \left[\max(\hat{d}(\cdot)) - \hat{d}(i)\right],$$

*where $\hat{d}(i)$ is the degree value at point $\boldsymbol{x}_i$, $\sigma^2$ is the bandwidth of the kernel $\boldsymbol{K}$ (Eq. 1) and $\boldsymbol{\Sigma}_i$ is the covariance of the Gaussian designed for generating new points (as described in Algorithm 1).*

In practice we suggest to use the mean of the upper and lower bound to set the number of generated points $\hat{\ell}(i)$. In Sec. 5.2 we demonstrate how the proposed scheme enables density equalization using few iterations of SUGAR. The proof of Prop. 4.1 is presented in the supplemental material.

## 5 Experimental Results

### 5.1 MNIST Manifold

In the following experiment we empirically demonstrate the ability of SUGAR to fill in missing samples and compare it to two generative Neural Networks: a Variational Autoencoder (VAE, Kingma & Welling, 2014), which has an implicit probabilistic model of the data, and a Generative Adversarial Network (GAN, Goodfellow et al., 2014), which learns to mimic input data distributions. Note that we are not able to use other density estimates in general due to the high dimensionality of datasets and the inability of density estimates to scale to high dimensions. To begin, we rotated an example of a handwritten '6' from the MNIST dataset in $N = 320$ different angles non-uniformly sampled over the range $[0, 2\pi]$. This circular construction was recovered by the diffusion maps embedding of the data, with points towards the undersampled regions having a lower degree than other regions of the embedding (Fig. 1, left, colored by degree). We then generated new points around each sample in the rotated data according to Alg. 1. We show the results of SUGAR before and after diffusion in Fig. 1 (top and bottom right, respectively).

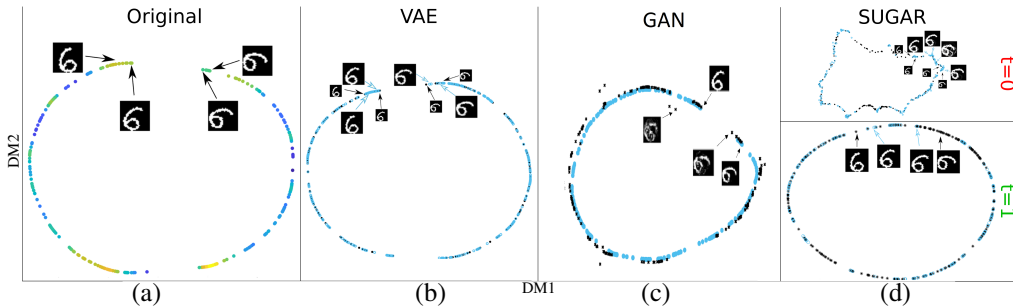

Figure 1: A two-dimensional DM representation of: (a) Original data, 320 rotated images of handwritten '6' colored by the degree value $\hat{d}(i)$. (b) VAE output; (c) GAN output; (d) Top: SUGAR augmented data before diffusion (i.e., $t = 0$); (d) Bottom: SUGAR augmented data with one step of diffusion ($t = 1$). Black asterisks – original data; Blue circles – output data.

Next, we compared our results to a two-layer VAE and a GAN trained over the original data (Fig. 1, (b) and (c)). Training a GAN on a dataset with number of samples of the same order as the dimension was not a simple task. Based on our experience adding the gradient penalty as suggested in Gulrajani et al. (2017) helps prevent mode collapse. The GAN was injected with uniform noise. Both SUGAR ($t = 1$) and the VAE generated points along the circular structure of the original manifold. For the

output of the GAN, we had to filter out around $5\%$ of the points, which fall far from the original manifold and look very noisy. Examples of images from both techniques are presented in Fig. 1. Notably, the VAE generated images similar to the original angle distribution, such that sparse regions of the manifold were not filled. In contrast, points generated by SUGAR occupied new angles not present in the original data but clearly present along the circular manifold. This example illustrates the ability of SUGAR to recover sparse areas of a data manifold.

## 5.2 Density Equalization

Given the circular manifold recovered in Sec. 5.1, we next sought to evaluate the density equalization properties proposed in Sec. 4.3. We begin by sampling one hundred points from a circle such that the highest density is at the origin ($\theta = 0$) and the density decreases away from it (Fig. 2(a), colored by degree $\hat{d}(i)$). SUGAR was then used to generate new points based on $\hat{\ell}(i)$ around each original point (Fig. 2(b), before diffusion, 2(c), after diffusion). We repeat this process for different initial densities and evaluate the resulting distribution of point against the amount of iteration of SUGAR. We perform a Kolmogorov-Smirnov (K-S) test to determine if the points came from a uniform distribution. The resulting p-values are presented in Fig. 2(d).

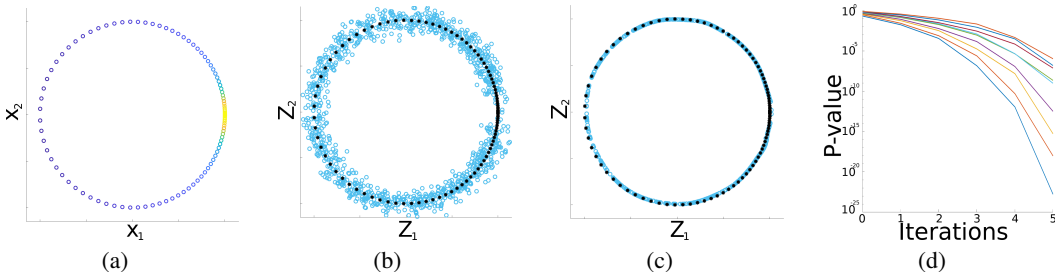

(a)  (b)  (c)  (d)

Figure 2: Density equalization demonstrated on a circle shaped manifold. (a) The original non-uniform samples of $X$. (b) The original points $X$ (black asterisks) and set of new generated points $Y_0$ (blue circles). (c) The final set of points $Z$, original points $X$ (black asterisks) and set of new generated points $Y_t$ (blue circles). In this example only one diffusion time step is required (i.e. $t = 1$). (d) The p-values of a Kolmogorov-Smirnov (K-S) test comparing to a uniform distribution, the x-axis represents the number of SUGAR iterations ran.

## 5.3 Classification of Imbalanced Data

The loss functions of many standard classification algorithms are global; these algorithms are thus easily biased when trained on imbalanced datasets. Imbalanced training typically manifests in poor classification of rare samples. These rare samples are often important (Weiss, 2004). For example, the preponderance of healthy individuals in medical data can obscure the diagnosis of rare diseases.

Resampling and boosting strategies have been used to combat data imbalance. Removing points (undersampling) is a simple solution, but this strategy leads to information loss and can decrease generalization performance. RUSBoost (Seiffert et al., 2010) combines this approach with boosting, a technique that resamples the data using a set of weights learned by iterative training. Oversampling methods remove class imbalance by generating synthetic data alongside the original data. Synthetic

Table 1: Average class precision (ACP), class recall (ACR), and the Matthews correlation coefficient (MCC) for k-NN and kernel SVM classifiers (using 10-fold cross validation) before / after SMOTE and SUGAR, and for RUSBoost classification.

|  | k-NN | | | SVM | | | RUSBoost |
|---|---|---|---|---|---|---|---|
|  | Orig | SMOTE | SUGAR | Orig | SMOTE | SUGAR | |
| ACP | 0.67 | 0.76 | **0.78** | 0.77 | 0.77 | **0.78** | 0.75 |
| ACR | 0.64 | 0.73 | **0.77** | 0.78 | 0.78 | **0.84** | 0.81 |
| MCC | 0.66 | 0.74 | **0.78** | 0.78 | 0.78 | **0.84** | 0.80 |

Minority Over-sampling Technique (SMOTE, Chawla et al., 2002) oversamples by generating points along lines between existing points of minority classes.

We compared SUGAR, RUSBoost, and SMOTE for improving k-NN and kernel SVM classification of 61 imbalanced datasets of varying size (from hundreds to thousands) and imbalance ratio (1.8–130), obtained from Alcalá-Fdez et al. (2009). To quantify classification performance we used Precision, Recall, and the Mathews correlation coefficient (MCC), which capture classification accuracy in light of data imbalance. For binary classification, precision measures the fraction of true positives to false positives, recall measures the fraction of true positives identified, and MCC is a discrete version of Pearson correlation between the observed and predicted class labels. Formally, they are defined as

$$\text{Precision} = \frac{TP}{TP + FP} \qquad \text{Recall} = \frac{TP}{TP + FN}$$

$$\text{MCC} = \frac{TP \cdot TN - FP \cdot FN}{\sqrt{(TP + FP)(TP + FN)(TN + FP)(TN + FN)}} \ .$$

For handling multiple classes, the first two are extended via average class precision and recall (ACP and ACR), which are defined as

$$\text{ACP} = \frac{1}{C} \sum_{c=1}^{C} \text{Precision}(class = c) \qquad \text{ACR} = \frac{1}{C} \sum_{c=1}^{C} \text{Recall}(class = c) \ ,$$

while MCC is extended to multiclass settings as defined in Gorodkin (2004). These metrics ignore class population biases by equally weighting classes. This experiment is summarized in Table 1 (see supplement for full details).

### 5.4 Clustering of Imbalanced Data

In order to examine the effect of SUGAR on clustering, we performed spectral clustering on a set of Gaussians in the shape of the word "SUGAR" (top panel, Fig. 3(a)). Next, we altered the mixtures to sample heavily towards points on the edges of the word (middle panel, Fig. 3(a)). This perturbation disrupted letter clusters. Finally, we performed SUGAR on the biased data to recreate data along the manifold. The combined data and its resultant clustering is shown in the bottom panel of Fig. 3(a) revealing that the letter clustering was restored after SUGAR.

The effect of sample density on spectral clustering is evident in the eigendecomposition of the graph Laplacian, which describes the connectivity of the graph and is the basis for spectral clustering. We shall focus on the multiplicity of the zero eigenvalue, which corresponds to the number of connected components of a graph. In our example, we see that the zero eigenvalue for the ground truth and SUGAR graphs has a multiplicity of $5$ whereas the corrupted graph only has a multiplicity of $4$ (see Fig. 3(b)). This connectivity difference arises from the $k$-neighborhoods of points in each ground

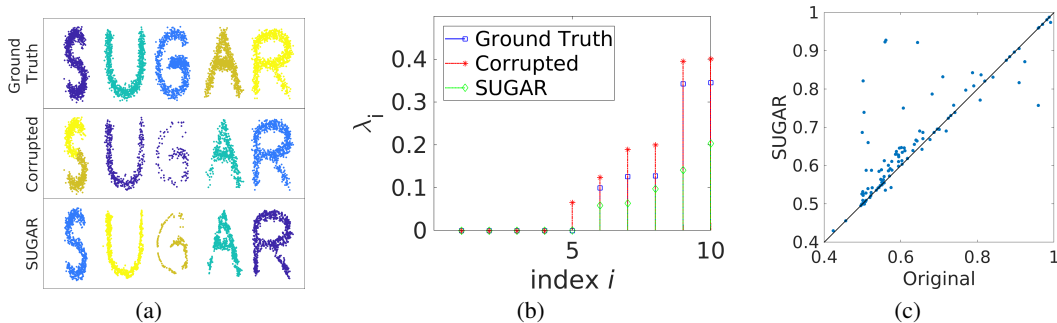

Figure 3: Augmented clustering using SUGAR. (a) Spectral clustering of a mixed Gaussian (top panel) with uneven sample density (middle panel). After SUGAR (bottom panel), the original cluster geometries are recovered. (b) Graph Laplacian eigenvalues from (a); the corrupted graph has a lower multiplicity of the zero eigenvalue, indicating fewer connected components. (c) Rand Index of 115 data sets (from Alcalá-Fdez et al., 2009) clustered by k-means before and after applying SUGAR.

truth cluster. We note that variation in sample density disrupts the $k$-neighborhood of points in the downsampled region to include points outside of their ground truth cluster. These connections across the letters of "SUGAR" thus lead to a lower multiplicity of the zero eigenvalue, which negatively affects the spectral clustering. Augmenting the biased data via SUGAR equalizes the sampling density, restoring ground-truth neighborhood structure to the graph built on the data.

Next, we explored the effects of SUGAR on traditional k-means across 115 datasets obtained from Alcalá-Fdez et al. (2009). K-means was performed using the ground truth number of clusters, and the Rand Index (RI, Hubert & Arabie, 1985) between the ground truth clustering and the empirical clustering was taken (Fig. 3(c), x-axis). Subsequently, SUGAR was used to generate new points for clustering together with the original data. The RI over the original data was again computed, this time using the SUGAR clusters (Fig. 3(c), y-axis). Our results indicate the SUGAR can be used to improve the cluster quality of k-means.

## 5.5 Biological Manifolds

Next, we used SUGAR for exploratory analysis of a biological dataset. In Velten et al. (2017), a high dimensional yet small ($\mathbf{X} \in \mathbb{R}^{1029 \times 12553}$) single-cell RNA sequencing (scRNA-seq) dataset was

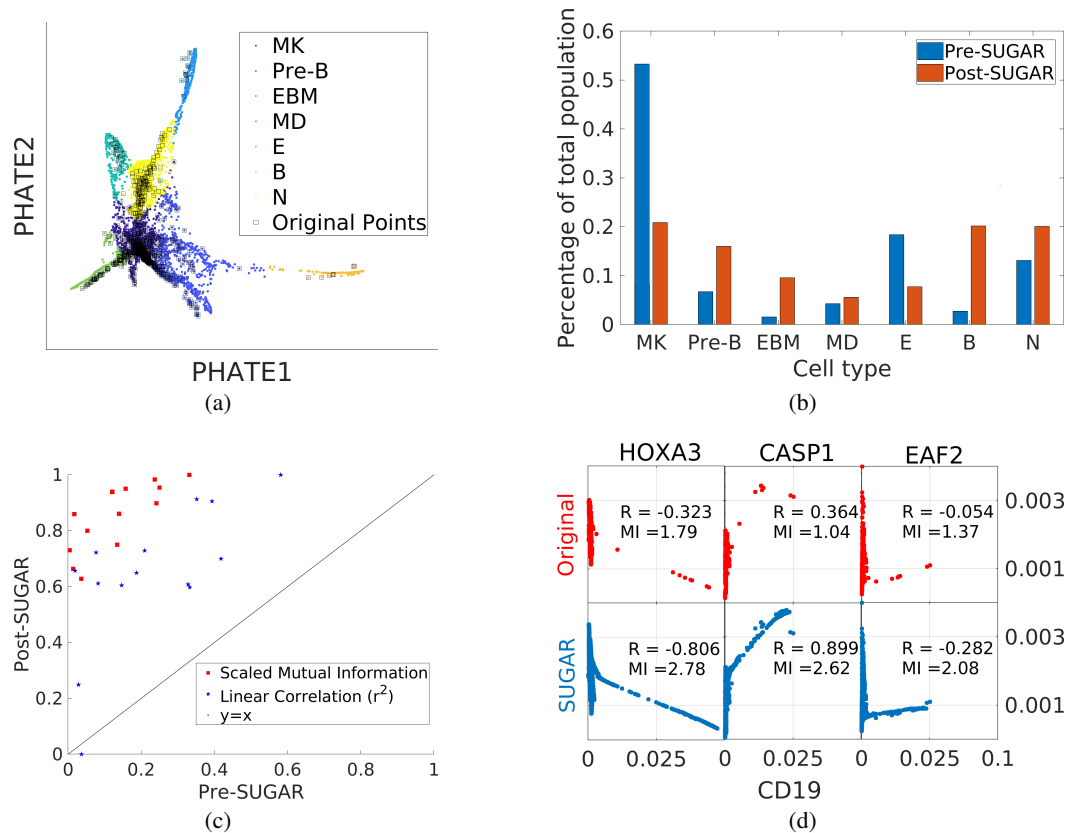

Figure 4: SUGAR was used to augment scRNA-seq data collected by Velten et al. (2017). (a) Augmented data embedded with PHATE (Moon et al., 2017b) and colored by k-means over the gene module dimensions identified by Velten et al. (2017). Seven canonical cell types are present. EBM: eosinophil/basophil/mast cells; N: neutrophils; MD: monocytes/dendritic cells; E: erythroid cells; MK: megakaryocytes; Pre-B: immature B cell; B: mature B cell. (b) Cell type prevalence before and after SUGAR. (c) Explained variation ($r^2$) and scaled mutual information ($\frac{MI_i}{\max MI}$) between the components of the fourteen coexpression modules identified by Velten et al. (2017). (d) Relationship between CD19 (B cell maturation marker) and HOXA3 (a cell immaturity marker), CASP1 (B cell linear commitment marker), and EAF2 (neutrophil and monocyte marker that is upregulated in mature B cells; see Sec.5.5). Marker names appear above the figure, values represented on the $y$ axis.

collected to elucidate the development of human blood cells, which is posited to form a continuum of development trajectories from a central reservoir of immature cells. This dataset thus represents an ideal substrate to explore manifold learning (Moon et al., 2017a). However, the data presents two distinct challenge due to 1. undersampling of cell types, and 2. dropout and artifacts associated with scRNA-seq (Kim et al., 2015). These challenges stand at odds with a central task of computational biology; namely, the characterization of gene-gene interactions that foment phenotypes.

We first sought to enrich rare phenotypes in the Velten data by generating $\mathbf{Y} \in \mathbb{R}^{4116 \times 12553}$ new data points with SUGAR. A useful tool for this analysis is the 'gene module', a pair or set of genes that are expressed together to drive phenotype development. K-means clustering of the augmented data over fourteen principal gene modules (32 dimensions) revealed six cell types described in Velten et al. (2017) and a seventh cluster consisting of mature B-cells (Fig. 4(a)). Analysis of population prevalence before and after SUGAR revealed a dramatic enrichment of mature B and pre-B cells, eosinophil/basophil/mast cells (EBM), and neutrophils (N), while previously dominant megakaryocytes (MK) became a more equal portion of the post-SUGAR population (Fig. 4(b)). These results demonstrate the ability of SUGAR to balance population prevalence along a data manifold.

In Fig. 4(c), we examine the effect of SUGAR on intra-module relationships. Because expression of genes in a module are molecularly linked, intra-module relationships should be strong in the absence of sampling biases and experimental noise. After SUGAR, we note an improvement in linear regression ($r^2$) and scaled mutual information coefficients. We note that in some cases the change in mutual information was stronger than linear regression, likely due to nonlinearities in the module relationship. Because this experiment was based on putative intra-module relationships we next sought to identify strong improvements in regression coefficients *de novo*. To this end, we compared the relationship of the B cell maturation marker CD19 with the entire dataset before and after SUGAR. In Fig. 4(d) we show three relationships with marked improvement from the original data (top panel) to the augmented data (bottom panel). The markers uncovered by this search, HOXA3, CASP1, and EAF2, each have disparate relationships with CD19. HOXA3 marks stem cell immaturity, and is negatively correlated with CD19. In contrast, CASP1 is known to mark commitment to the B cell lineage (Velten et al., 2017). After SUGAR, both of these relationships were enhanced. EAF2 is a part of a module that is expressed during early development of neutrophils and monocytes; we observe that its correlation and mutual information with B cell maturation are also increased after SUGAR. We note that in light of the early development discussed by Velten et al. (2017), this new relationship seems problematic. In fact, Li et al. (2016) showed that EAF2 is upregulated in mature B cells as a mechanism against autoimmunity. Taken together, our analyses show that SUGAR is effective for bolstering relationships between dimensions in the absence of prior knowledge for exploratory data analysis.

## 6   Conclusion

SUGAR presents a new type of generative model, based on data geometry rather than density. This enables us to compensate for sparsity and heavily biased sampling in many data types of interest, especially biomedical data. We assume that the training data lies on a low-dimensional manifold. The manifold assumption is usually valid in many datasets (e.g., single cell RNA sequencing (Moon et al., 2017a)) as they are globally high-dimensional but locally generated by a small number of factors. We use a diffusion kernel to capture the manifold structure. Then, we randomly generate new points along the incomplete manifold, with emphasis on its sparse areas. Finally, we use a weighted transition kernel to pull the new points towards the structure of the manifold. The presented method demonstrated promising results on synthetic data, MNIST images, and high dimensional biological datasets in applications such as clustering, classification, and mutual information relationship analysis. We note that a toolbox implementing the presented algorithm is available via GitHub[3] for free academic use (see supplement for details), and we expect future work to apply SUGAR to study extremely biased biological datasets and improve classification and regression performance on them.

**Acknowledgments**

This research was partially funded by grant from the Chan-Zuckerberg Initiative (ID: 182702).

## Footnotes

[3]URL: github.com/KrishnaswamyLab/SUGAR

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
