[Supplementary Material]

# Supplemental Material

**Ofir Lindenbaum**[*]
Applied Mathematics Program
Yale University
New Haven, CT 06511
`ofir.lindenbaum@yale.edu`

**Jay S. Stanley III**[*]
Computational Biology & Bioinformatics Program
Yale University
New Haven, CT 06510
`jay.stanley@yale.edu`

**Guy Wolf**[†]
Applied Mathematics Program
Yale University
New Haven, CT 06511
`guy.wolf@yale.edu`

**Smita Krishnaswamy**[†] ⊠
Departments of Genetics & Computer Science
Yale University
New Haven, CT 06510
`smita.krishnawamy@yale.edu`

## 1 Proof of proposition 4.1

*Proof.* Given $\boldsymbol{x}_i \in \mathbb{R}^D, i = 1, ..., N$, the degree $\hat{d}(i)$ at point $\boldsymbol{x}_i$ is defined as discussed in Section 4.2. SUGAR (Algorithm 1) generates $\hat{\ell}(i)$ points around $\boldsymbol{x}_i$ from a Gaussian distribution $\mathcal{N}(\boldsymbol{x}_i, \boldsymbol{\Sigma})$. After generating $\hat{\ell}(i)$ points, the degree at point $\boldsymbol{x}_i$ can be defined as

$$\bar{d}(i) = \sum_{j=1}^{N} e^{-\frac{\|\boldsymbol{x}_i - \boldsymbol{x}_j\|^2}{2\sigma^2}} + \sum_{\ell=1}^{\hat{\ell}(i)} e^{-\frac{\|\boldsymbol{x}_i - \boldsymbol{y}_\ell^i\|^2}{2\sigma^2}} + \sum_{\tilde{\ell}} e^{-\frac{\|\boldsymbol{x}_i - \boldsymbol{y}_{\tilde{\ell}}\|^2}{2\sigma^2}}. \tag{1}$$

In order to equalize the distribution, we require that the expectation of the degree be independent of $\boldsymbol{x}_i$, thus, we set $\mathbb{E}\left[\bar{d}(i)\right] = C, i = 1, ..., N$, where C is a constant that will be addressed later in the proof. Further, we notice that the first term on the right hand side of Eq. 1 can be substituted by $\hat{d}(i)$, while the second term is a sum of $\hat{\ell}(i)$ random variables. The third term accounts for the influence of points generated around $\boldsymbol{x}_j, j \neq i$ on the degree at point $\boldsymbol{x}_i$. While it is hard to find a closed form solution for the number of points required at each $i$, we can derive bounds on the required value based on two simple assumptions. First, to find an upper bound, we assume that the points generated around $\boldsymbol{x}_i$ have negligible effect on the $\hat{d}(j)$ at point $j \neq i$. This leads to an upper bound, because in practice the degree is affected by other points as well. We note that this assumption should hold with proper choices of $\sigma$ and $\boldsymbol{\Sigma}_i, i = 1, ..., N$. By substituting in the constant C, the term $\hat{d}(i)$, and using the independence assumption, the expectation of Eq. 2 can be written as

$$C - \hat{d}(i) = \mathbb{E}\left[ \sum_{\ell=1}^{\hat{\ell}(i)} e^{-\frac{\left\|\boldsymbol{x}_i - \boldsymbol{y}_\ell^i\right\|^2}{2\sigma^2}} \right]. \tag{2}$$

Then, since the variables $\boldsymbol{y}_\ell^i$ are i.i.d., then the right hand side of Eq. 2 becomes

$$\hat{\ell}(i) \, \mathbb{E}\left[ e^{-\frac{\left\|\boldsymbol{x}_i - \boldsymbol{y}^i\right\|^2}{2\sigma^2}} \right] =$$

$$\frac{\hat{\ell}(i)}{\det\left(2\pi\boldsymbol{\Sigma}_i\right)^{1/2}} \int_{\mathbb{R}^D} e^{-\frac{\left\|\boldsymbol{x}_i - \boldsymbol{y}^i\right\|^2}{2\sigma^2}} e^{-(\boldsymbol{x}_i - \boldsymbol{y}^i)^T \boldsymbol{\Sigma}_i^{-1} (\boldsymbol{x}_i - \boldsymbol{y}^i)} d\boldsymbol{y}^i,$$

---

[*]These authors contributed equally
[†]These authors contributed equally; ⊠ Corresponding author

and then using a change of variables $\tilde{\boldsymbol{y}} = \boldsymbol{y}^i - \boldsymbol{x}_i$, the integral simplifies to

$$\frac{\hat{\ell}(i)}{\det\left(2\pi\boldsymbol{\Sigma}_i\right)^{1/2}} \int_{\mathbb{R}^D} e^{-(\boldsymbol{x}_i - \boldsymbol{y}^i)^T \left(\boldsymbol{\Sigma}_i^{-1} + \frac{\boldsymbol{I}}{2\sigma^2}\right)(\boldsymbol{x}_i - \boldsymbol{y}^i)} d\boldsymbol{y}^i =$$

$$\frac{\hat{\ell}(i)}{\det\left(\boldsymbol{I} + \frac{\det(\boldsymbol{\Sigma}_i)}{2\sigma^2}\right)^{0.5}},$$

where the final step is based on the integral of a Gaussian and a change of variables. Finally, this leads to an upper bound $\hat{\ell}(i) \leq \det\left(\boldsymbol{I} + \frac{\det(\boldsymbol{\Sigma}_i)}{2\sigma^2}\right)^{0.5}[C - \hat{d}(i)]$ on the number of points $\hat{\ell}(i)$. By choosing $C = \max(\hat{d}(\cdot))$ we guarantee that $\hat{\ell}(i) \geq 0$, which means we are not removing points.

Now, by considering the third term on the right side of Eq. 1, we derive a lower bound on $\hat{\ell}(i)$. The term $\sum_{\tilde{\ell}} e^{-\frac{\|\boldsymbol{x}_i - \boldsymbol{y}_{\tilde{\ell}}\|^2}{2\sigma^2}}$ accounts for points generated around neighbor points around $\boldsymbol{x}_i$. The number of points in the region of $\|\boldsymbol{x}_i - \boldsymbol{x}_j\|^2 \lesssim \sigma^2$ is proportional to the degree $\hat{d}(i)$. For the lower bound, we assume that the number of points generated by each neighbor of $\boldsymbol{x}_i$ is no more than $\hat{\ell}(i) + 1$. This assumption becomes realistic by imposing smoothness on the function $\hat{\ell}(i)$. Plugging this assumption and the degree value as a measure of connectivity into Eq. 1 we get that

$$\bar{d}(i) \leq \hat{d}(i) + \sum_{\ell=1}^{\hat{\ell}(i)} e^{-\frac{\|\boldsymbol{x}_i - \boldsymbol{y}_\ell^i\|^2}{2\sigma^2}} + \hat{d}(i) \sum_{\tilde{\ell}=1}^{\hat{\ell}(i)+1} e^{-\frac{\|\boldsymbol{x}_i - \boldsymbol{y}_{\tilde{\ell}}\|^2}{2\sigma^2}},$$

we now take the expectation of both sides

$$C \leq \hat{d}(i) + \mathbb{E}\left[\sum_{\ell=1}^{\hat{\ell}(i)} e^{-\frac{\|\boldsymbol{x}_i - \boldsymbol{y}_{\tilde{\ell}}^i\|^2}{2\sigma^2}}\right] + \hat{d}(i)\,\mathbb{E}\left[\sum_{\tilde{\ell}=1}^{\hat{\ell}(i)+1} e^{-\frac{\|\boldsymbol{x}_i - \boldsymbol{y}_{\tilde{\ell}}\|^2}{2\sigma^2}}\right]$$

$$\leq \hat{d}(i) + \left[\hat{d}(i) + 1\right]\mathbb{E}\left[\sum_{\ell=1}^{\hat{\ell}(i)+1} e^{-\frac{\|\boldsymbol{x}_i - \boldsymbol{y}_\ell^i\|^2}{2\sigma^2}}\right],$$

by replacing $\boldsymbol{y}_{\tilde{\ell}}$ by $\boldsymbol{y}_\ell^i$ we increase the expectation value. Finally, exploiting the i.i.d. assumption and the Gaussian distribution we conclude that by replacing the mean with similar steps as used for the upper bound we conclude that

$$\det\left(\boldsymbol{I} + \frac{\boldsymbol{\Sigma}_i}{2\sigma^2}\right)^{0.5}\left[\max(\hat{d}(\cdot)) - \hat{d}(i)\right] / \left[\hat{d}(i) + 1\right] - 1 \leq \hat{\ell}(i),$$

which completes the proof. $\qquad\square$

## 2 Implementation

SUGAR has been implemented in Matlab and in Python. The results presented in Section 5 are based on the Matlab implementation. The code is accessible at: github.com/KrishnaswamyLab/SUGAR. The implementation follows Algorithm 1, with modification to allow flexibility in choosing the number of points $\ell$ and estimating density. We remark that the main hyper-parameter $k$ should be set based on the connectivity of existing data points.