[Reviews · NeurIPS 2018]

Reviewer 1



Geometry based data generation This paper proposes to a geometric method to generate data by exploiting the intrinsic shape of the data manifold. This contrasts with typical approaches where data density is estimated, rather than directly exploiting the geometry of the data manifold. The methodology builds upon the diffusion maps of Coifman & Lafon but uses a different kernel. Taken separately, diffusion maps and the MGC kernel are not new. However, the method, SUGAR, can elegantly estimate the intrinsic geometry of a data manifold, particularly in heavily unbalanced datasets, and consequently generate synthetic data in undersampled area. This has high potentials in a series of applications where data is severely imbalanced. Experiments are performed on synthetic data, such as with rotate MNIST digits, and real data, where relationships between gene-phenotype is studied. Results are convincing and clearly demonstrate the benefit of exploiting the intrinsic geometry of the data manifold. On a possible negative side, the proposed method depends on a k-nearest neighbor search - this is a general question, but it remains unclear how to choose k, and how to prevent having disconnected components. This may imply additional assumptions on the data manifold, for instance, not guaranteeing a connected manifold. Authors indicated that no k-nn has been used, instead a Gaussian kernel provides a diffusion process across the graph

Reviewer 2



The authors introduce here an innovative generative model for data, based on learning the manifold geometry rather than the density of the data. They claim that this technique can deal w/ sparsity and heavily biased sampling. In practice, this is realised through a diffusion kernel which generates new points along the incomplete manifold by randomly generating points at sparse areas. The paper is well written, with a grounded theoretical frameworks and based on an interesting novel idea. Although no ingredient of the pipeline could be considered really novel, the overall solution can be deemed original because of the proposed smart combination of the components. The experimental section is very rich and all the results are consistently supporting all the authors' claims; in particular, the results obtained on biological datasets are biologically plausible and meaningful, making the proposed method particularly suitable for this kind of data. A number of points can be raised whose discussion might enhance the manuscript value: - what is the relation with the stochastic block model in clustering? - what is the relation with UMAP and/or tSNE in dimensionality reduction and manifold approximation? - I would strongly suggest using MCC as the performance metric ## After reading the authors' rebuttal, I confirm my rating: the observations included in the rebuttal are convincing and further supporting my evaluation of the manuscript as a good submission.

Reviewer 3



It is the difficult to discover small clusters from imbalanced dataset in classification/clustering tasks. The authors try to tackle this problem by imputing new data points into the original dataset using a diffusion map based approach. Diffusion map is a Gaussian kernel normalized by a diagonal matrix D, whose diagonal elements are the total connectivity of the corresponding data point. This kernel can be further extended to measure-based Gaussian Correlation (MGC) kernel, which only considers a set of reference points. The main idea is to generate new data points using the MGC kernel in neighbourhood of each original data point and combine them for downstream classification/clustering tasks. The authors then discusses how to select the bandwidth of the Gaussian kernel, how many new data points to generate for each original data point, how to set the number of multiplication steps of the Gaussian kernel. After that the authors provide a proof about setting l(i) which I do not go through. The MNIST experiment shows the method is able to interpolate data points in low density regions, while variational autoencoder is not. After that the authors check the generated data points are uniformly distributed in the manifold. Then they show SUGAR achieves better performance compared with other data sampling methods on classification and clustering of imbalanced datasets. In the final single cell dataset, they show SUGAR helps to separate different cell types from the data. In general I feel the key strength of SUGAR comes from the random walk near the original data points, which may help to connect the low density regions. The exploration property comes from the diffusion distance (line 141). The larger t used in the diffusion distance, the closer the newly generated data points to the original. While solving the minor cluster problem, I have a gut feeling imputing new data points may bring up new problems such as fake new clusters. I have some questions regarding the experiments: 1. MINST: this experiments may be too simple. Only one "6" is rotated to generate the whole dataset, so there is no noise in principle. Comparison with GAN instead of VAE may be more realistic. In Figure 1, how is t determined? 2. Clustering of imbalanced data: Figure 3a shows the contribution of SUGAR is merging the two clusters in "S" but not imputing the low density regions in "G" (bottom part), what is the t used here? How is K chosen in k-means for the newly generated dataset? 3. Biological data: Figure 4d, what are the x-axis and y-axis? 4. what is "f" in line 133? Although with flaws, I feel this is a nice paper to be accepted. Updates: Based on the authors' feedback, it seems to be a nice paper.